# Structure-Aware Trail Bundling for Large DTI Datasets

**Steven Bouma[1]** , **Christophe Hurter [2]** **and Alexandru Telea [3],***

1  Bernoulli Institute, University of Groningen, 9747AG Groningen, The Netherlands; stevenbouma@gmail.com
2  French Civil Aviation Center, University of Toulouse, 31055 Toulouse, France; christophe.hurter@enac.fr
3  Department of Information and Computing Science, Utrecht University, 3655CC Utrecht, The Netherlands
*  Correspondence: a.c.telea@uu.nl; Tel.: +31-30-253-4170

**Abstract:** Creating simplified visualizations of large 3D trail sets with limited occlusion and preservation of the main structures in the data is challenging. We address this challenge for the specific context of 3D fiber trails created by DTI tractography. For this, we propose to jointly simplify trails in both the geometric space (by extending and adapting an existing bundling method to handle 3D trails) and in the image space (by proposing several shading and rendering techniques). Our method can handle 3D datasets of hundreds of thousands of trails at interactive rate, has parameters for the most of which good preset values are given, and produces visualizations that have been found, in a small-scale user study involving five medical professionals, to be better in occlusion reduction, conveying the connectivity structure of the brain, and overall clarity than existing methods for the same data. We demonstrate our technique with several real-world public DTI datasets.

**Keywords:** trail bundling; DTI visualization; tractography

## 1. Introduction

Trails or trajectories are generated by many applications, such as graph drawing and graph visualization [1], monitoring of vehicle motion along large geographical spaces [2,3] and scanning techniques [4–6]. Besides their spatial position, trails can be annotated by additional data attributes, such as weights, directions, and types of the entire trail, or speed and travel duration of the item traveling along the trail. Visualizing datasets consisting of hundreds of thousands of trails is challenging, due to the high levels of occlusion, and the difficulty of simultaneously displaying multiple attributes.

In information visualization, *bundling* has emerged as an important technique to handle such datasets [7]. *Edge bundling* creates simplified visualizations of large graphs by deforming the (typically straight-line) drawing of spatially close and data-similar edges so these get visually grouped, thereby creating whitespace that reduces clutter and allows one to discern the main structure of the graph drawing. *Trail bundling* generalizes the process to arbitrary sets of curves, or trails, embedded in 2D space. Numerous bundling algorithms have been proposed in the last two decades, offering a wealth of drawing styles, heuristics for grouping the edges or trails, and acceleration strategies to handle datasets of hundreds of thousands of elements in real time [8].

Although bundling has proven its added value for 2D graph-and-trail simplification, only a handful of works have explored its generalization to 3D datasets. Bundling graphs laid out on 3D surfaces has been met with success [9]. However, for the graph visualization use-case, truly volumetric 3D layouts are far less common [1]. The other data type—trails—is also approached by only a few

methods. This leaves the question whether 3D trail bundling is an efficient and effective tool for the visual exploration of such data.

A salient use-case for 3D bundling is the visual exploration of Diffusion Tensor Imaging (DTI) datasets via tractography [10–12]. This application generates massive sets of 3D curves, representing the paths followed by major fiber tracts in the brain. Due to the volumetric structure of the brain and its complex connection patterns, obtaining simplified views of such patterns has raised much interest in the visualization community. This has been approaches by two main method types. First, so-called *illustrative* rendering methods aim to draw the actual, raw, 3D fiber tract data in ways that make important spatial patterns show up more saliently [13–15]. These methods work well for medium-density trail sets; too-high-density ones create, inevitably, occlusions that do not allow visual exploration of a full volume. Secondly, *3D bundling* methods take the more aggressive step of actually deforming the trails to create more structure at the expense of preserving spatial information [16]. To our knowledge, these methods cannot handle large 3D datasets consisting of hundreds of thousands of trails. Moreover, the *combination* of illustrative techniques and 3D bundling techniques has been explored only to very limited extents. This is in stark contrast to the aforementioned results for bundling-and-rendering techniques for 2D trail sets [8].

In this paper, we aim to answer the question of how to combine *bundling* and *rendering* techniques efficiently and effectively for the visual simplification of large DTI fiber tracts. For this, we adapt a state-of-the-art method for 2D trail bundling [8] to efficiently handle 3D DTI trail sets, taking into account the typical structures that exist and need to be emphasized in such data. Next, we propose a range of rendering techniques to present the 3D bundling results so that the structures of interest in the data get further emphasized. Our joint 3D-bundling-and-rendering pipeline can handle large 3D fiber tract datasets of hundreds of thousands of trails in real time on consumer-grade hardware; offers several exploration modes that highlight different aspects of the DTI tract data; and allows various ways to control the process leading to simplified views of the data. We demonstrate our method on several real-world DTI datasets. We also present a qualitative user study of our method by five medical professionals that outlines several advantages in terms of occlusion reduction, conveying the connectivity structure of the brain, and overall clarity of the produced visualizations.

The structure of this paper is as follows. Section 2 discusses related work. Section 3 details our method. Section 4 discusses our implementation. Sections 5 and 6 present two evaluations of our method. Section 7 discusses our main findings. Finally, Section 8 concludes the paper.

## 2. Related Work

Related work can be classified in the following two main topics.

### 2.1. Diffusion Tensor Imaging and Tractography

Conventional structural imaging techniques such as T1-, T2- and proton density-weighted imaging generally create high contrast between major tissue groups in the brain, which are: Gray Matter (GM), White Matter (WM), and Cerebrospinal Fluid (CSF). Such structural imaging techniques are generally well suited for the study of tissue macrostructure yet provide little insight into the orientation of white matter fibers. DWI [17,18] is a variant of conventional Magnetic Resonance Imaging (MRI) based on the tissue water diffusion rate, which is better suited for the study of white matter pathways. Although DWI refers to the contrast of the acquired images, DTI is a specific type of modeling (or abstraction) of the DWI datasets, in which diffusivity of water molecules is represented by tensors [10]. As an in-vivo non-destructive technique that requires no chemical tracers, DTI is presently one of the most promising methods for the study of white matter architecture in living humans. DTI provides quantitative estimates of white matter integrity and orientation by measuring molecular diffusion of water molecules (or Brownian motion). It is based on the phenomenon of diffusion anisotropy in the nerve tissue: water molecules diffuse faster along the neural fiber direction and slower in the fiber-transverse direction.

DTI tensor datasets record, essentially, the spectrum of diffusion strengths per diffusion direction at every point in a 3D scan. Such datasets can be visualized using glyph techniques, which render the local major, medium, and minor eigenvalues of the diffusion tensor, scaled by their corresponding eigenvalues, over a dense sampling of the 3D scanned volume [29]. *Tractography* methods improve in the above by reducing occlusion and also explicitly showing the fiber tract paths by essentially integrating streamlines in the major direction of the DTI eigenvector, seeded from a suitably chosen point set [11,12]. More elaborate approaches include streamlines defined by tensor deflection [10], bi-tensor modeling [19] and streamsurface tracking [20]. Fiber tracking has several applications, including noninvasive visualization of white matter pathways, segmenting of specific tracts in the brain for image analysis, and relating white matter tract anatomy to brain tumors and lesions in patients who are candidates for neurosurgery [10].

Tractography essentially delivers large sets of (hundreds of thousands of) 3D fiber trails embedded in the 3D scan volume. Visualizing these next is challenging. Conventional line rendering is a simple and efficient baseline rendering technique to show such datasets that has been employed since fiber tracking was first introduced [12,21]. Yet, such naive rendering of the densely seeded and criss-crossing 3D trails creates too much clutter and occlusion for many tasks to be accomplished effectively. Numerous visualization tools have been proposed to leverage interactivity to explore large DTI datasets, e.g., [22]. However, navigating the large space of parameter settings can be daunting for users. More extensive illumination [23], ambient occlusion [24] and alpha-blending [25] have been applied to line rendering of fiber tracts to further improve their visual exploration. Rendering 3D tubes with Phong shading [26] is based on more detailed geometric modeling than line rendering, and can produce even higher quality visualizations. The methods by Stoll et al. [27] and Merhof et al. [28] produce similar shading using screenspace techniques and incorporate illustrative rendering techniques [15] as well. Hyperstreamlines [25,29] are an extension of cylindrical tubes that provide a richer representation of the DTI field.

Clustering forms separate class of methods that help visualizing large tract sets. Such methods detect trails that are similar in terms of spatial position, diffusion values, and possibly other attributes (e.g., curvature, length) and group them into clusters. These in turn enable one to create a simplified visualization by e.g., rendering each cluster with a different color or reducing the cluster to a simpler representative, which is next visualized. Xu et al. [30] use the DBSCAN clustering algorithm [31] to create such clusters and combine them with user specification of regions of interest to create a rich palette of focus-and-context fiber tract visualization. Poco et al. [32] approach this by reducing every 3D fiber to a high-dimensional feature vector that represents position, geometry, and smoothness. Next, dimensionality reduction [33] is used to create 2D scatterplots where similar fibers appear as point clusters. This enables one to easily select similar-fiber bundles by simply selecting point clusters in the projection. Comparisons of fiber clustering methods are presented in [34,35]. Interestingly, some clustering methods, such as DBSCAN or mean shift [36], are related to density estimation, which is also used by CUBu [8], the bundling technique that we adapt for our DTI visualization (see further Section 2.2). The key difference between our method and clustering is that we *deform* the implicitly clustered fibers to simplify the resulting visualization.

Surface reconstruction or extraction from DTI datasets have also been applied as a means for indirect volume visualization [29]. Merhof et al. [28] have shown that encompassing fiber tract bundles with isosurfaces yields a preferable representation for use in neurosurgery. Ridge and valley surfaces [29] are demonstrated to capture the cores of sheet-like tracts. Visualizing such surfaces provides important added value compared to all fiber-based alternatives. Indeed, even if there is a dense surface-like distribution of fiber tracts, creating the appearance of a *surface* from this (without gaps) can be challenging when applying line-rendering-based techniques. Besides explicit surface extraction, such surfaces can be also created *implicitly* in the image space. The Depth-Dependent Halos (DDH) method [13,14,37] performs this merging of structures quite well, as colinear fibers are visually combined into thicker bundles with illustrative halos. The major downside of DDH is that

the black-and-white rendering (a stylistic choice motivated by nonphotorealistic rendering) offers limited scalar visualization capabilities. Our method, discussed next, can reproduce DDH but also add shading and information color-coding to the fiber rendering.

### 2.2. Trail and Edge-Bundling Techniques

In contrast to the above-mentioned rendering techniques, which modify how a 3D trail set is *depicted*, trail and edge-bundling techniques modify the actual *trail set* to emphasize structures of interest. Bundling essentially aims to (a) identify trails that are spatially close and similar from the perspective of one or several data attributes; and (b) deform these trails so they get spatially closer, so as to create more visual structure in the ensuing rendering thereof. Applications of edge bundling include graph drawing simplification [8,38], trajectory exploration [7], eye-tracking analysis [5] and streamline bundling [39].

Bundling of 2D datasets has been extensively explored. Image-based edge-bundling techniques, including SBEB [40], ADEB [5], KDEEB [38], CUBu [8], and FFTEB [41] show that the problem of bundling of 2D trails, as well as the rendering of resulting bundles using a variety of styles, can be efficiently and effectively addressed using image processing approaches that scale well on graphics hardware. Bundling is however far less explored for 3D volumetric trail datasets such as DTI tracts. Bottger et al. [42,43] have proposed mean-shift bundling of 3D connectivity graphs obtained through functional Magnetic Resonance Imaging (fMRI). Simplification of anatomical connectivity using edge bundling is a relatively new topic, one example thereof being KDEEB [38] applied to fiber tracts [25,44]. Other examples of fiber tract simplification methods include multi-scale local fiber tract contraction [16] and two-dimensional neural maps [45]. To our knowledge, such 3D methods have two main limitations: They cannot, in the same time (1) recover well plausible anatomical structures, such as the mix of surfaces and tube-like structures present in a DTI dataset; and (2) allow interactive visual exploration of DTI datasets of hundreds of thousands of fibers in real time using different rendering styles. For example, the method in [16], which is technically closest to ours, requires hours to complete on a dataset of 78 K fiber tracts. At a higher level, bundling-and-rendering methods for DTI data have evolved largely separated, with the notable exception of [16]. This, we believe, is an artificial separation of two method classes which ultimately aim to the same goal—presenting a large, complex, dense, volumetric 3D trail dataset in a *simplified* manner to the user.

Summarizing the above, our contribution to the creation of simplified visualization of large DTI trail sets is three-fold:

1. We present a method for bundling 3D DTI tracts that allows the user to preserve relevant underlying anatomical elements (fiber tubular bundles, sheets, and manifolds) in a controlled manner;
2. We propose several rendering styles of the 3D bundled structures that emphasize several aspects of interest in the data;
3. We propose a joint implementation of the above two points that can handle 3D trail datasets of hundreds of thousands of trails in real time on consumer graphics hardware.

As our method builds atop of the backbone of the CUBu technique [8]. We inherit from CUBu the kernel density-based advection of trails that ultimately creates the bundles, and the full GPU-based bundling computation for computational scalability. As described next, we modify CUBu in several respects: We implement the entire pipeline in 3D (using a volumetric density map, Section 3.2); allow trail endpoints to bundle under specific constraints (Section 3.2.1); use the DTI volume anisotropy rather than the CUBu isotropic bundling (Section 3.2.2); reseed tracts in sparsely populated volume areas rather than bundling a fixed, predefined, trail set (Section 3.2.3); and propose several rendering modes designed to emphasize structure specific to DTI fiber sheets (Section 3.3).

### 3. Method

To describe our joint rendering-and-bundling method for 3D DTI tracts, we first introduce some notations. Let $V : \mathbb{R}^3 \rightarrow \mathbb{R}^6$ be a 3D volume scan that encodes, per voxel, the symmetric 3 by 3 DTI diffusion tensor. For every point $\mathbf{x} \in V$, we can compute the three eigenvectors $\mathbf{e}_i$ and eigenvalues $\lambda_i$, $1 \le i \le 3$, of $V$'s diffusion matrix, which give the directions, respectively strengths, of the largest, medium, and smallest diffusions recorded by the MRI data[46]. Our visualization method then consists of three steps: seeding and tracing DTI trails (Section 3.1); bundling trails to follow anatomical structures (Section 3.2); and structure-emphasizing rendering (Section 3.3).

*3.1. Seeding and Tracing DTI Trails*

To start with, we need to construct the actual 3D trails representing streamlines in the major DTI eigenvector $\mathbf{e}_1$. The standard method to seed such streamlines is to look at regions $M \subset V$ of so-called *high anisotropy*, i.e., points $\mathbf{x} \in V$ for which the diffusion is far stronger in one direction than in all others. $M$ can be computed by finding high-anisotropy voxels in $V$, i.e., points where the eigenvalues $\lambda_1, \ldots, \lambda_3$ strongly differ[46]; alternatively, $M$ is readily provided by so-called *masks*, suitably segmented from $V$ by medical professionals based on their knowledge of fiber-rich versions[47]. The masks $M$ we use in all our work are of the latter type, and they are provided with the public DTI datasets we use in our experiments[48–50]. Figure 1 shows such a mask for one of the datasets used in our work.

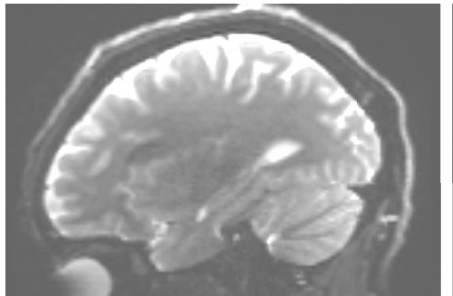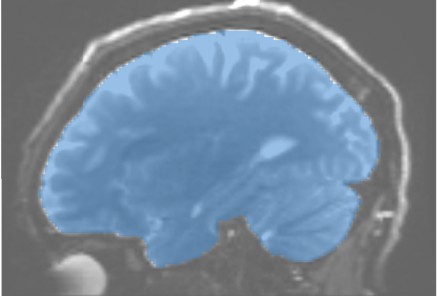

**Figure 1. Left**: Sagittal slide from a DWI volume. **Right**: Mask (in light blue) for the brain region within this volume. See Section 3.1.

We use such regions $M$ for two purposes. First, we densely seed $M$ to *start* tracing streamlines in the vector field $\mathbf{e}_1$ from points inside it. Secondly, we *stop* tracing such streamlines when they exit $M$, therefore avoiding that they enter low-anisotropy regions. Given a dense seed-set of points $S \subset M$, we trace a streamline $\mathbf{t}_i$ in the vector field $\mathbf{e}_1$ for each seed point $\mathbf{x}_i \in S$ until $\mathbf{t}_i$ exists $M$. The set of streamlines $T = \{\mathbf{t}_i\}$ represents our dense 3D tract set we aim next to simplify (Section 3.2) and render (Section 3.3).

*3.2. DTI Bundling*

As outlined in Section 2.2, the aim of DTI bundling is to produce a visually simpler, easier to understand, structure of the dense trail set $T$ in which salient tubes and manifolds (surfaces) inherently present in the brain-fiber structure show up. For this, we adapt the original isotropic CUBu bundling method in[8], as follows. To explain this, we first briefly introduce CUBu (for full details, we refer to[8]). For ease of reading, we reuse below, where applicable, the same notations as in[8].

**Resampling:** For CUBu to work well, we need $T$ to be densely, and uniformly, sampled. Given a user-specified sampling step $\sigma$, we resample all trails $\mathbf{t}_i \in T$ so that the average distance between consecutive sampling points $\mathbf{x}_j, \mathbf{x}_{j+1}$ on a trail $\mathbf{t}_i$ equals $\sigma$.

**Density computation:** Given our trail set $T$, we first compute a 3D density map $\rho$ for points $\mathbf{x}_j$ uniformly sampled along the 3D trails $\mathbf{t}_i \in T$ as

$$\rho(\mathbf{x}) = \sum_{\mathbf{x}_j \in T, \|\mathbf{x}-\mathbf{x}_j\| \leq P_R} K(\|\mathbf{x}-\mathbf{x}_j\|) \tag{1}$$

where $K : \mathbb{R}^+ \to \mathbb{R}^+$ is a parabolic (Epanechnikov) kernel of width $P_R$, shown to yield better kernel density estimates (KDEs) than e.g., Gaussian kernels [38]. The kernel width $P_R$ (in voxels) controls the visualization's simplification level but also the sample points $\mathbf{x}_j$ considered when evaluating Equation (1). Typical good values are $P_R$ in 5% to 10% of the extent of the volume $V$. Figure 2 shows the effect of $P_R$ upon the bundling. We see that increasing $P_R$ simplifies the 3D trail set $T$ more. Similar effects have been reported for 2D trail bundling applications [8], but not yet demonstrated for the bundling of 3D trail sets.

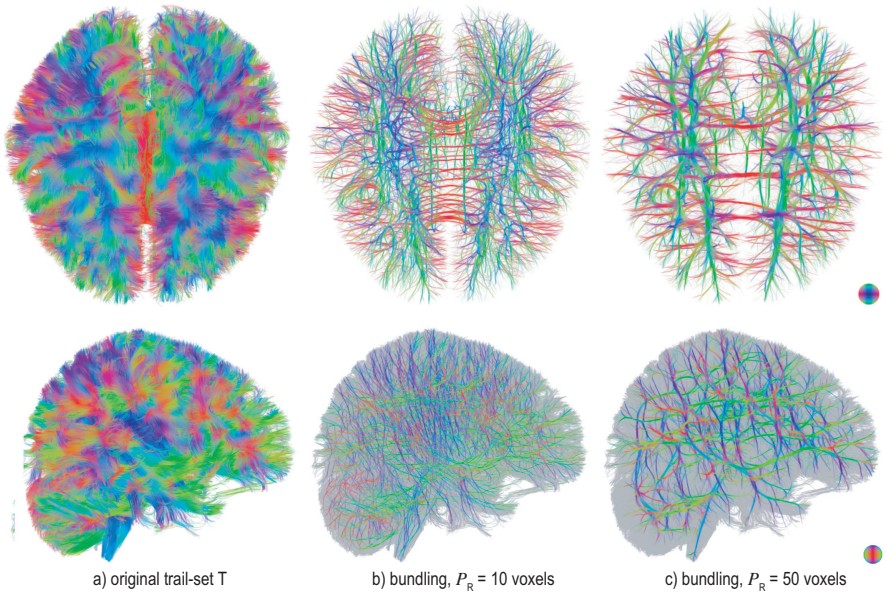

a) original trail-set T      b) bundling, $P_R$ = 10 voxels      c) bundling, $P_R$ = 50 voxels

**Figure 2.** Effect of kernel-width bundling parameter $P_R$. Rows show two different views of the same dataset. Trails are directionally colored according to the displayed legend. Gray shows the seeding-and-trail-clipping mask $M$. See Section 3.2.

**Advection:** We next use the density map $\rho$ to advect every trail-sampling-point $\mathbf{x}_j$ in the normalized density gradient, following

$$\mathbf{x}_j^{new} = \mathbf{x}_j + P_R \frac{\nabla \rho(\mathbf{x}_j)}{\|\nabla \rho \mathbf{x}_j\|}. \tag{2}$$

**Smoothing:** Next, following CUBu [8], we smooth the advected trails $T^{new} = \{\mathbf{x}_j^{new}\}$ by applying a 1D Laplacian kernel of size $L$ sampling points and strength $\phi$ (see Equation (3)) to remove potential noise effects caused by the discrete advection in Equation (2).

$$\mathbf{x}_j^{smooth} = (1-\phi)\mathbf{x}_j^{new} + \frac{\phi}{2L+1} \sum_{k=j-L}^{j+L} \mathbf{x}_k^{new} \tag{3}$$

The smoothing kernel size $L$ is set so as to match the KDE kernel radius $P_R$, i.e., $\|\mathbf{x}_j - \mathbf{x}_{j-L}\| \approx P_R$.

**Relaxation:** Finally, we apply a relaxation step that interpolates linearly between the bundled trails $B(T) = \{\mathbf{x}_j^{smooth}\}$ and the original ones $T$ as $T^{relax} = \gamma B(T) + (1-\gamma)T$. Figure 3 shows this effect for two different $\gamma$ values. Simply put, $\gamma$ interpolates between a fully bundled (tight) visualization $B(T)$

($\gamma = 1$) and a fully relaxed visualization showing the original DTI trail set $T$ ($\gamma = 0$). Using the preset $\gamma = 0.2$ gave good results for all datasets we experimented with.

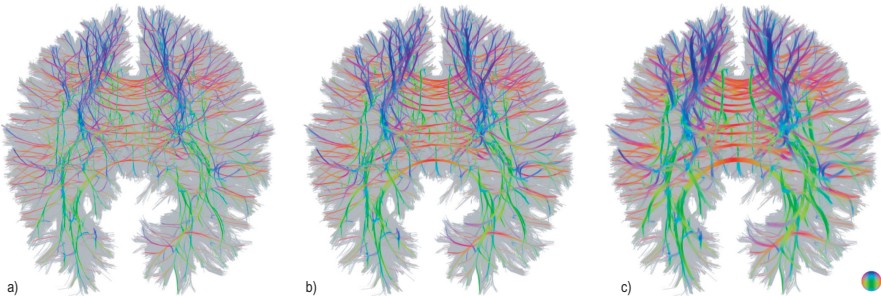

**Figure 3.** Effect of relaxation parameter $\gamma$. (**a**) Unbundled dataset $T$. (**b**) Bundled dataset, $\gamma = 0.1$. (**c**) Bundled dataset, $\gamma = 0.2$. Rendering parameters are the same as in Figure 2. See Section 3.2.

We repeat all above steps (relaxation, resampling, density computation, advection, smoothing, and resampling) $P_N$ times. As shown in [8] for 2D trail sets, and verified by us for 3D trails, the process converges yielding strongly bundled trails after $P_N = 15$ iterations.

### 3.2.1. Constrained Endpoint Advection

The original CUBu bundling [8], applied to 2D trails, keeps trail *endpoints* fixed (not affected by Equation (2)) and updates all other trail sample points $x_j$. This is well motivated by the fact that CUBu (and, actually, for all other bundling methods we are aware of) address 2D trails where endpoints represent *meaningful* information, such as nodes in a graph layout or fixation points in a 2D eye-tracking dataset [7]. In contrast, our 3D trail endpoints are *far less* meaningful, as they are simply locations of low confidence for the streamline tracing process (Section 3.1). Hence, their location does not encode any significant information. Keeping them fixed creates fan-like structures that generate significant occlusion, preventing us to look into the 3D bundle structure. As such, we choose to advect these trail endpoints, just as all other sampling points $x_j$.

Naively applying Equation (2) to endpoints however has the effect of *shortening* the fiber tracts, therefore losing important spatial information. To alleviate this, we constrain the advection $\mathbf{d} = x_j^{new} - x_j$ to be normal to the local tangent $\tau = x_{j+1} - x_j$ of the fiber at sample point $x_j$. Let $\mathbf{d}^\perp$ be the constrained advection vector. Please note that for 2D trail sets $T$, the direction $\mathbf{u}$ of $\mathbf{d}^\perp$ can be directly found as the cross product $\tau \times \mathbf{v}$, where $\mathbf{v}$ is the normal vector to the 2D plane containing the 2D trail set $T$ [8]. In 3D, we have an infinity of such directions $\mathbf{u}$ orthogonal to $\tau$. We solve the problem of computing this orthogonal direction $\mathbf{u}$ and the constrained advection $\mathbf{d}^\perp$ in direction $\mathbf{u}$ as follows (see also Figure 4a):

$$\mathbf{d}^\perp = \begin{cases} 0, & \text{if } \frac{\mathbf{d}}{\|\mathbf{d}\|} \cdot \tau \approx 1 \\ \mathbf{d}, & \text{if } \frac{\mathbf{d}}{\|\mathbf{d}\|} \cdot \tau \approx 0 \\ (\mathbf{d} \cdot \mathbf{u}) \cdot \mathbf{u}, & \text{where } \mathbf{u} = \tau \times (\mathbf{d} \times \tau), \text{ otherwise.} \end{cases} \tag{4}$$

Figure 4b–d shows the effect on trail endpoints of this constrained advection. Image (b) shows an unbundled trail set $T$ with 50 K fibers seeded and traced to cover the entire brain. As visible, it is hard to see structures inside the brain due to occlusion. Image (c) shows the classical, unconstrained, bundling delivered by directly applying Equation (2). We see how fiber bundles form inside the brain. However, fiber endpoints are blocked, so they create fan-like structures that still severely limit visibility inside the structure. Image (d) shows our constrained bundling obtained by Equation (4). The terminal fan-like structures now largely disappear allowing us to peep inside the bundled structure with limited occlusion trouble.

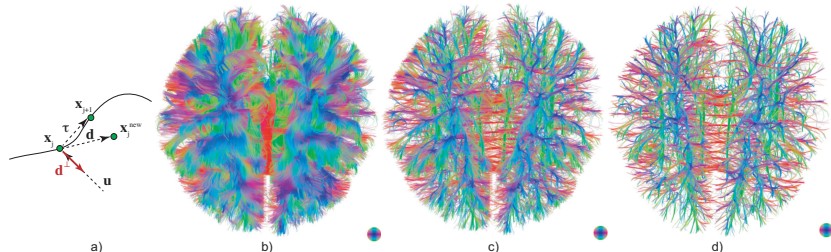

**Figure 4.** (**a**) Constrained advection scheme for trail endpoints. (**b**) Unbundled fiber set. (**c**) Unconstrained bundling. (**d**) Bundling with constrained advection.

### 3.2.2. Anisotropy-Constrained Bundling

Figure 4d shows that 3D bundling considerably decreases occlusion, allowing one to see structures created by fibers deep inside the trail set. However, a problem immediately appears: All manifold-like structures, such as known to exist in DTI fiber tracts, are now broken into a set of parallel bundles, separated by gaps. This is a serious problem as the resulting rendering does not faithfully convey the brain anatomy, as also reported in other works that use DTI fiber contraction [16]. Figure 5 shows this. Image (a) shows an unbundled trail set. Given that this is a dense dataset (100 K trails), the manifold structure of the *corpus callosum* is well captured, albeit occluded by the outer fiber tracks. Image (b) shows the isotropic bundling computed by the constrained advection explained earlier (Section 3.2.1). Occlusion is massively reduced, but the *corpus callosum* manifold is broken into a set of bundles separated by gaps.

We address this problem by modulating the advection, i.e., replacing the advection step $\mathbf{d}^{\perp}$ (Equation (4)) by $\kappa \mathbf{d}^{\perp}$, where

$$\kappa(\mathbf{x}_j) = \begin{cases} 1, & \text{if } \alpha(\mathbf{x}_j) \geq \alpha_0 \\ 0, & \text{otherwise,} \end{cases} \tag{5}$$

where $\alpha(\mathbf{x}_j)$ is any suitable DTI anisotropy metric computed at sample point $\mathbf{x}_j$ and $\alpha_0$ is a threshold thereof. Figure 5c shows the effect of using fractional anisotropy (FA) [51] for $\alpha$, defined as

$$FA = \sqrt{\frac{1}{2}} \frac{\sqrt{(\lambda_1 - \lambda_2)^2 + (\lambda_2 - \lambda_3)^2 + (\lambda_3 - \lambda_1)^2}}{\sqrt{\lambda_1^2 + \lambda_2^2 + \lambda_3^2}}, \tag{6}$$

where $\lambda_i$ are the major, medium, and minor eigenvalues of the diffusion tensor evaluated at the current sample point $\mathbf{x}_j$. Figure 5c shows the effect of constraining bundling by *FA*. Fibers with a value $FA > \alpha_0$, where $\alpha_0 = 0.7$, are in regions of strong anisotropy indicating tube-like structures, and are as such strongly bundled. These are color-coded in the image by their *FA* values. *FA* values below $\alpha_0$ indicate planar anisotropy or isotropic regions, and are as such not bundled. These fibers are marked gray in the image. Image (d) shows the anisotropic bundling, color-cored directionally, for an easier comparison with the isotropic bundling (image (b)). As visible, anisotropic bundling reduces occlusion *and* also keeps manifold-like structures unbroken. Similar results can be obtained by using the $c_l$ and $c_p$ anisotropy metrics of Westin et al. [46], as shown further by the examples in Section 5.

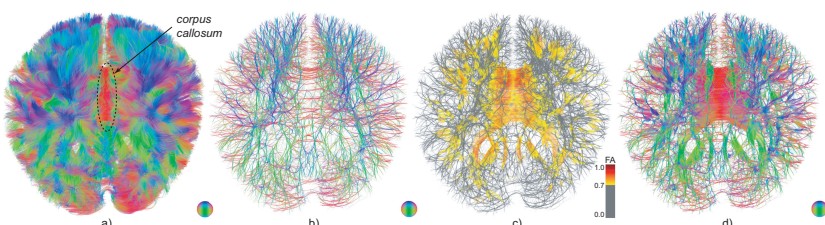

**Figure 5.** Anisotropic bundling. (**a**) Unbundled trail set. (**b**) Isotropic bundling. (**c**) Anisotropic bundling color-coded by *FA* metric. (**d**) As (**c**), with directional color coding. See Section 3.2.2.

### 3.2.3. Tract Reseeding

Anisotropic bundling (Section 3.2.2) cannot *fully* prevent the appearance of gaps in fiber manifolds. Indeed, if we set the anisotropic threshold $\alpha_0$ too high, then gaps will be prevented, since only the highest anisotropy fiber fragments, in linear anisotropy regions, will be bundled. However, this causes too little bundling, thus does not decrease occlusion sufficiently. Conversely, of we set $\alpha_0$ too low, then occlusion is strongly reduced, but bundling may also occur in planar anisotropy regions, creating the aforementioned gaps. Tuning $\alpha_0$ to strike the right balance between occlusion reduction and fiber manifold preservation is delicate. Moreover, such manifolds may not be fully captured by the original trail set $T$, unless a very high seed density is used, which only increases occlusion.

We solve this problem by *adaptive reseeding*, performed during the bundling process, as follows. After each bundling iteration, we create (seed) new trails in regions of high planar anisotropy $c_p$ and low linear anisotropy $c_l$, where

$$c_p = \frac{2(\lambda_2 - \lambda_3)}{\lambda_1 + \lambda_2 + \lambda_3} \tag{7}$$

and

$$c_l = \frac{\lambda_1 - \lambda_2}{\lambda_1 + \lambda_2 + \lambda_3}. \tag{8}$$

Similar to Vilanova et al. [20], we use the constraints $c_p \geq 0.25$ and $c_l \leq 0.2$ to find the seeding region, and trace fibers, randomly seeded at voxels in these regions, until they exit them, i.e., not meet the above-mentioned constraints. Fiber tracking is done using Euler integration along the major eigenvector of the DTI tensor, and trilinear interpolation. Figure 6 shows the effect of reseeding of a relatively small 15 K trail set. Reseeding adds a total of 65 K trails (marked in red) to the original 15 K ones (marked in gray).

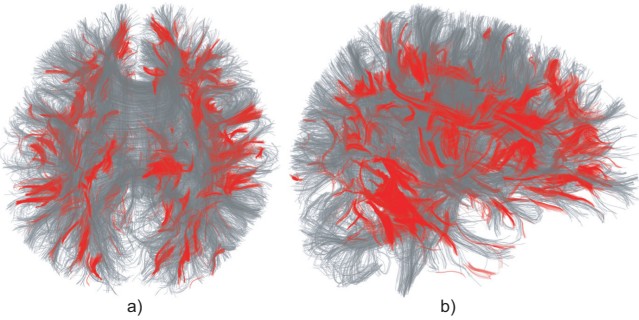

a)         b)

**Figure 6.** Fiber reseeding during anisotropic bundling, (**a**) anterior and (**b**) transversal views. Original fibers are in gray. Fibers added by reseeding are red. See Section 3.2.3.

### 3.3. Structure-Emphasizing Rendering

The additions to the classical CUBu pipeline discussed in Section 3.2—constrained advection, anisotropic advection, and reseeding—generate a bundled trail set $B(T)$ that has less occlusion than the original set $T$ but captures well the underlying structures created by fibers. As outlined in Section 2.1, rendering $B(T)$ using classical methods such as illuminated tubes [25,27,52,53] is a simple and fast way to get insights in the bundled trail set. However, even for very dense trail sets, recreating the continuous nature of fiber manifolds from individually rendered tubes is very hard. Depth-dependent halos (DDH) [13,14,37] methods achieve this goal by merging close fibers in screen space. The downside of DDH methods is that they create a black-and-white rendering, which, albeit motivated from a minimalistic design viewpoint, offers limited freedom to visualize additional scalar data.

We address the above by several rendering techniques that work on either the original trail set $T$ or the bundled one $B(T)$, as follows.

### 3.3.1. Splat-Based Rendering

Since the trail set $T$ is densely sampled, we can create solid-looking geometry by rendering circular splats, also called billboarded point sprites [54], oriented parallel to the view plane and centered at the sample points $\mathbf{x}_i \in T$. Our technique is related to the work of Jalba et al. [55] that reconstruct 3D shapes from point-cloud representations of their surface skeletons and to other methods for surface splatting from point clouds [56]. However, our aim is to render a complex set of *fibers*—either individual ones or grouped to form fiber sheets; in contrast, the other splatting works mentioned above aim to reconstruct a single, watertight, *surface* from a point cloud. In our method, we use different splat profiles than the above works; effectively use blending to merge splats into smooth-looking surfaces; show data atop of the reconstructed images by color coding; and create outlines to better convey the separation between the complex DTI fiber structures.

Our rendering pipeline consists of four main steps: shading and blending, gap-filling, and creating outlines, as discussed next.

### 3.3.2. Shading and Blending

As outlined at the beginning of Section 3.3.1, we render $T$ by drawing a viewplane-parallel circular luminance splat texture $t$ centered at the location of every sample point $\mathbf{x}_i \in T$. Tuning the splat radius effectively 'merges' close trails even further than what bundling can do, creating visually compact shapes. Choosing different profiles for the texture $t$ allows obtaining rendering effects that range from flat-shaded-like to smooth-shaded surfaces and finally to specularly shaded tubes showing local fiber details. We discuss next four such profiles. To illustrate them, we made a selection of trails that emerge from the *corona radiata* and merge with those in the *corpus callosum*. These selected trails are shown in Figure 7a, overlaid atop of a sagittal slice encoding fractional anisotropy (FA), for context. Figure 7b–e show the four luminance textures corresponding to these profiles (topmost inset) and also as a 3D height plot of the luminance values (inset below) and how these profiles affect the rendering of the selected trails. The profiles are detailed further below.

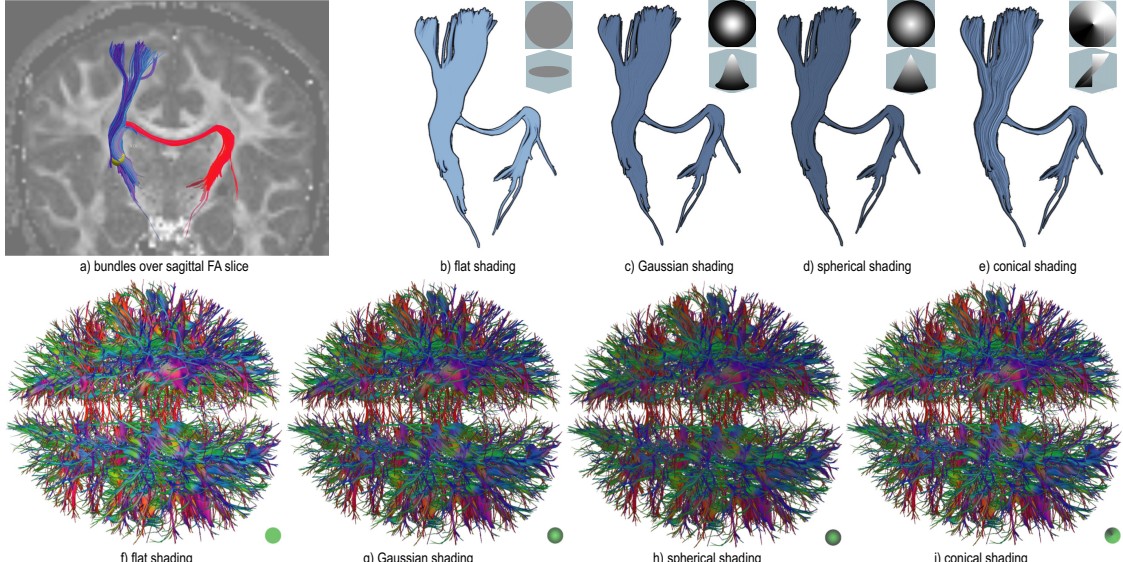

**Figure 7.** (**a**) Selected trail set consisting of a tube-like bundle from the *corona radiata* (blue) merging into the *corpus callosum* (red). (**b–e**) Same bundle as in (**a**) shaded with four different profiles. (**f–i**) The four shading profiles applied to a full 100 K-trails bundled dataset. See Section 3.3.2.

**Flat ($t_F$):** This profile contains a constant (bright) luminance and is useful when one wants to show additional information by e.g., color coding. When combined with outlining (discussed further in

Section 3.3.4), it shows a minimalistic, illustrative rendering style that allows easily separating trail bundles at a coarse scale. For instance, in Figure 7b, we easily see that the trail set consists of one large vertical bundle and one curved bundle—compare this image with Figure 7 where the same two bundles have been manually separated by color coding.

**Gaussian ($t_G$):** This profile sets the luminance texture to a Gaussian map—see the height-plot inset in Figure 7c. The actual luminance texture (also shown in the inset of Figure 7d) resembles a diffusely shaded sphere. The obtained effect is to create a subtle shading close to the borders of a bundle.

**Spherical ($t_S$):** This profile sets the luminance of a pixel $\mathbf{x} \in t$ to its distance to the circular splat's boundary. As visible in the insets in Figure 7d, the profile resembles a specularly shaded sphere; if we depict $t$ by a height plot, the profile shows as a conical shape. The obtained effect, also visible in Figure 7d, is to create slightly sharper highlights and shadows than the Gaussian profile.

**Conical ($t_C$):** This profile corresponds to a cone lit by a light source of direction $\mathbf{a}$, seen from above—see the insets in Figure 7e. We tested two options for setting the light direction $\mathbf{a}$, as follows (see also Figure 8). First, we use a global, fixed, direction $\mathbf{a}$. This is equivalent to the well-known directional lighting in OpenGL. Although easy to understand, and creating a granular rendering where individual trails can be discerned, fixed lighting does sometimes not create strong-enough shading cues to differentiate overlapping bundles (Figure 8a). Alternatively, we set $\mathbf{a}$ to a vector locally orthogonal to the view-space tangent $\tau$ to each trail, the latter being estimated by projecting the 3D line segment $(\mathbf{x}_j, \mathbf{x}_{j+1})$ to the view plane—see inset in Figure 8b. The effect is that lighting always 'sticks' to the same side of a trail, and follows the trail's curvature. As visible in Figure 8b, this creates stronger shading contrasts allowing one to separate easier overlapping bundles.

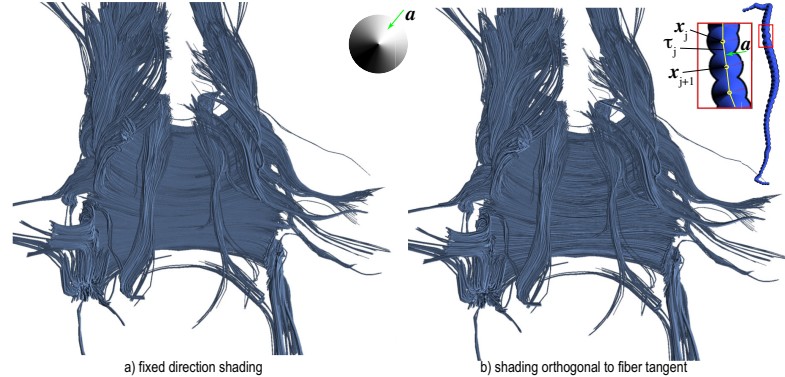

a) fixed direction shading                                                    b) shading orthogonal to fiber tangent

**Figure 8.** Conical shading with (**a**) fixed light direction and (**b**) lighting orthogonal to local fiber tangent. The latter method (**b**) shows more shading contrast than the former (**a**). See Section 3.3.2.

Figure 7f–i show the application of the above four shading profiles to a fill 100 K-trails bundled dataset. Here, we use color (hue) to encode the trails' directions, similar to earlier images in this paper (e.g., Figure 2). Combining color with shading is immediately done by multiplying the luminance values produced by the profile textures by the color values computed by mapping the trail directions at every sample point $\mathbf{x}_j$.

Directly splatting trail sets however creates artifacts where the borders of the circular splats overlap—see Figure 9a and its insets. A better solution is to use alpha-blending to render the splats. This would require depth-sorting all splats every time the viewpoint changes. For a dataset of hundreds of thousands of trails, each rendered with hundreds of splats, this is not (easily) done at interactive frame rates. Solutions such as Order-Independent Transparency (OIT)[57,58] do not require depth-sorting the splats and are as such faster. However, OIT works well typically for a few overlapping surfaces with high transparency. Our datasets, in contrast, contain hundreds of

overlapping fibers at a single location. More importantly, we do not need high-transparency rendering, but rather low-transparency, sufficient to allow the few overlapping splats in the 'topmost' layer, i.e., closest to the viewpoint, to blend. We achieve this by a simple two-pass rendering technique: First, we render the entire trail set $T$ using no shading but with Z buffering on. This delivers a depth buffer $\mathbf{z}$ recording the $z$ value of the closest splat to the viewplane at every pixel. In the second pass, we visit (again) all splats, but only render those whose depths, i.e., the $z$ components of the sample points $\mathbf{x}_j$ where they are centered, are not larger than the $\mathbf{z}$ value at that pixel plus a small offset $\delta = 0.005$. This effectively renders only those splats that are within a 'peel', or thin layer, of $\delta$ units from the closest one to the viewer. Since there are only a few splats per pixel within this layer—typically under 10—we can use standard blending with a transparency value of $\alpha = 0.1$. Figure 9b shows the same dataset as in Figure 9a rendered with peel blending. As visible, peel blending removes the shading artifacts we had seen earlier.

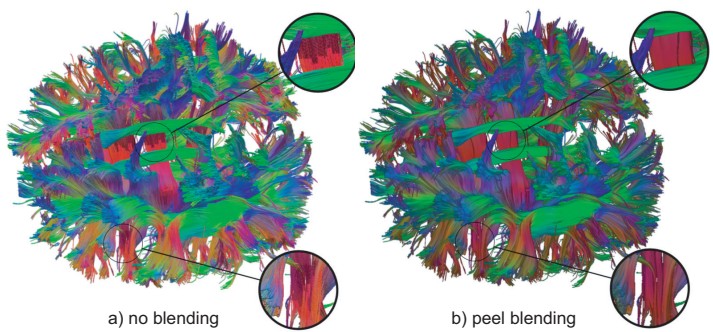

a) no blending                    b) peel blending

**Figure 9.** Trail set visualized with splatting with (**a**) no blending and (**b**) peel blending. Note how blending removes artifacts. See Section 3.3.2.

### 3.3.3. Gap Filling

The key added value of splatting is to fill gaps between close-and-parallel trails in a bundle so as to make it appear as a compact, smooth, structure—thereby removing the inherently discrete, sampled, nature of trail sets. To do this, we need to carefully set the radius of the splat textures $t$ so that splats are not too large (in screen space) so that they fill in large gaps between far-apart bundles, which we want to see as being distinct, and in the same time they are not too small, so that splats at consecutive sampling points $\mathbf{x}_j$ on a trail do not overlap sufficiently so as to give the impression of a constant-thickness tube. Additionally, we want that splats that render fiber sheets are larger so that gaps in such sheets are visually closed. Finally, we want to make fibers far from the viewer look thinner to account for perspective.

We achieve the above by computing the radius $r$ of the splat rendered at sample point $\mathbf{x}_j$ as

$$r(\mathbf{x}_j) = (r_0 + c_p(\mathbf{x}_j)\beta)\sqrt{\frac{1}{z_j + z_j^2}} \tag{9}$$

where $r_0$ is the base splat radius (set typically to around 5 pixels, so as to ensure that all fibers are sufficiently visible); $c_p(\mathbf{x}_j)$ is the planar anisotropy at location $\mathbf{x}_j$; $\beta$ is a constant factor determining how much we increase splats in planar regions; and $z_j$ is the depth value of $\mathbf{x}_j$ after perspective transformation. Please note that the square-root term in Equation (9) is similar to the well-known distance-based attenuation commonly used in lighting [59]. Figure 10a,b show the effect of the parameter $\beta$: For $\beta = 0$, we do not make splats in planar regions larger. As such, even though splats overlap in these regions, the individual fibers are still visible here (see insets in Figure 10a). Setting $\beta = 5$ makes splats in these regions larger, therefore allowing for more overlap, thus making the surface-like fiber sheets more apparent in planar regions (see insets in Figure 10b).

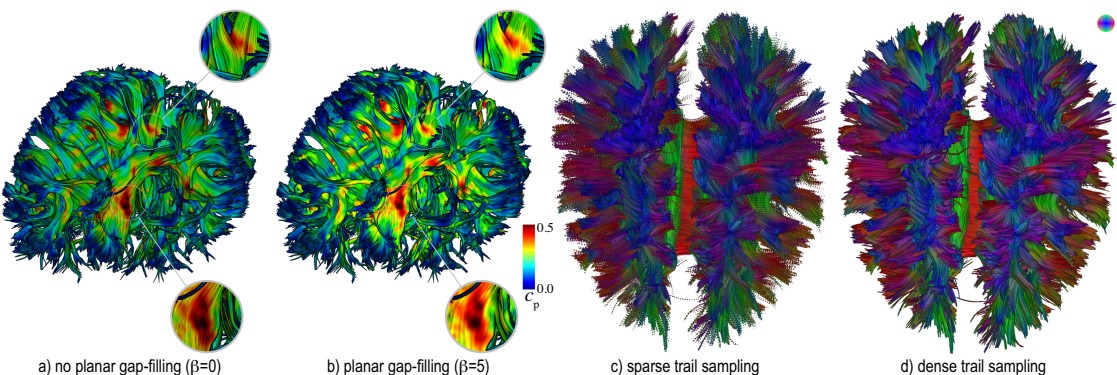

**Figure 10.** Trail set visualized (**a**) no emphasis of planar regions; (**b**) gap-filling in planar regions; (**c**) sparse, and (**d**) dense trail sampling. Images (**a**,**b**) are colored by planar anisotropy $c_p$. Images (**c**,**d**) are colored on trail direction. See Section 3.3.3.

Besides controlling the splat radius, we need to control the *density* of sample points $\mathbf{x}_j$ along a trail. Indeed: The original CUBu algorithm, which we extended to 3D (Section 3.2), creates a dense, and uniform, but *fixed* sampling of the trail set. Although this sampling density is typically sufficient for rendering edge bundles as polylines in typical graph drawings [8], it can be too low to ensure sufficient overlap of consecutive splats during blending (Section 3.3.2). Increasing the base radius $r_0$ (Equation (9)) can fill in such gaps, but it also makes trails look overall thicker, which may not be desirable when one wants to see fine details. Moreover, the trails sampling is fixed, meaning that gaps will inevitably appear when one zooms in to examine details. Figure 10c illustrates the above: Here, we use a small splat radius $r_0 = 3$ and the sampling distance $\|\mathbf{x}_j - \mathbf{x}_{j+1}\|$ to roughly 1 voxel. This creates clear gaps in the trails rendering. We solve this problem by applying on-the-fly tessellation to the line segments $(\mathbf{x}_j, \mathbf{x}_{j+1})$ as these are rendered to generate splat centers. As the user zooms in, such line segments are subdivided to generate more sample points, thereby ensuring a constant and high sample point density in image space. Figure 10d illustrates this: The density achieved here by tessellation is roughly 7 times larger than in Figure 10c, which ensures that consecutive splats blend seamlessly to create continuous-looking *and* thin, detail-rich, trails.

### 3.3.4. Creating Outlines

Our final rendering addition proposed involves the generation of *outlines* along the boundaries of trail bundles in image space. These outlines are useful to emphasize groups of close and roughly parallel trails, which helps one to disambiguate crossing or overlapping bundles, by e.g., seeing which bundle is in front of another one [13,60].

We can easily create outlines by slightly adapting our two-pass rendering method used for peel blending (Section 3.3.2 as follows. In the first pass, used to create the depth buffer $\mathbf{z}$, we render splats with a slightly larger radius $r + \epsilon$, where $r$ is the standard radius given by Equation (9). Then, all pixels that have a $\mathbf{z}$ value lower than the maximal one (that is, set when clearing the Z buffer) and that have a color in the framebuffer different from background—that is, pixels that are visited by the first rendering pass but not the second rendering pass—are on an outline of $\epsilon$ pixels thickness of rendered fibers. In practice, setting $\epsilon$ to a few pixels gives salient enough, but not too bold (visually disturbing) outlines. Figure 11 illustrates this: In the left set of images, outlines are those pixels that are purple in the Z buffer (rendered there) and yellow in the framebuffer (not rendered there). Image (b) in this figure shows the added value of outlines: Compared to standard shading, it is now easier to discern how bundles overlap and cross each other. For more clarity, we also show the outlines only in image (c). Separately, note that our outline-creation technique *subsumes* the DDH technique in [13]: We can create DDH-like images simply by enabling outlines (rendered in black) and using a constant splat profile with white base color (Section 3.3.2).

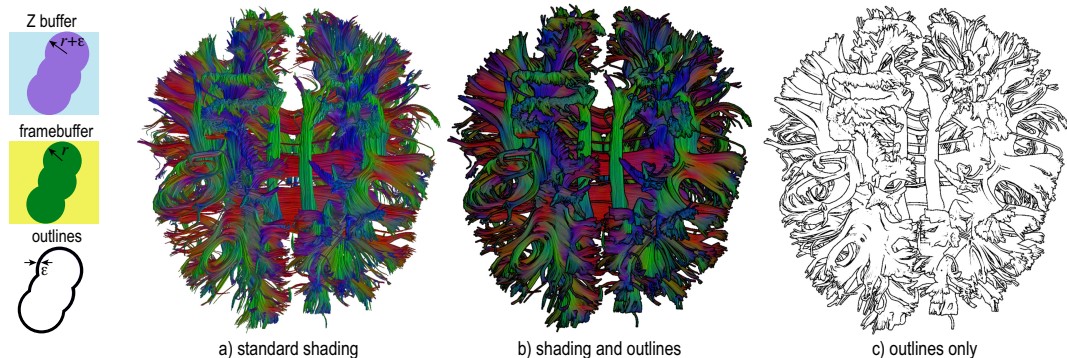

**Figure 11.** Creating bundle outlines. Trail set rendered (**a**) with shading but no outlines, (**b**) with shading and outlines, and (**c**) as outlines only. See Section 3.3.4.

## 4. Scalable Implementation

We implemented our end-to-end pipeline fully on the GPU using NVidia CUDA and programmable OpenGL (pixel shaders and tessellation) and C++ for the main architecture. The various scalar volumes (DTI, fractional anisotropy, density $\rho$) are implemented as 32-bit floating-point 3D textures. Density estimation (Equation (1)) is implemented by separable convolution requiring three one-dimensional passes. For the density map $\rho$, we use volumes of higher (up to 8 times) resolution than the original DTI volumes (Table 1), as the latter are too coarse for the accuracy needed to compute the gradient $\nabla \rho$ (Equation (2)). Gradient estimation is done by central differences.

The above implementation can efficiently compute and render the fiber bundling of large trail sets. Table 2 shows, for three different DTI datasets, the number of trails $|T|$, sampling points $\mathbf{x}_j$, average time per bundling iteration (over 15 total iterations), and total bundling-and-rendering time. Experiments were done on a PC running Ubuntu Linux using a NVidia GTX 1080 graphics card (2560 CUDA cores). Time includes the CPU-to-GPU transfer of trail sets needed to load the input dataset $T$ on the GPU prior to doing the bundling proper. The rendering time is a small fraction of the total time—under 4%—and as such not reported separately. For each of the three datasets, we constructed two trail sets, one with 50 K trails, and the other one with 500 K trails (all of them randomly seeded). Details over these available datasets are shown in Table 1. The table lists the (public) data sources, identification of the dataset in terms of patient (subject) ID and accession number, and data characteristics—DTI volume size, voxel size, and number of diffusion directions used to acquire the DTI tensor field. Summarizing Table 1, we argue that our data is compatible with real-world, large, and complex DTI volumes used in medical practice.

**Table 1.** Datasets used in the evaluation.

|  | **Dataset 1** | **Dataset 2** | **Dataset 3** |
|---|---|---|---|
| Data source | 3DSlicer [48] | OpenfMRI [49] | HCP [50] |
| Subject ID | n/a | Subject 10159 | mgh_1001 |
| Accession number | n/a | ds000030 | S01322 |
| Volume sizes | $128 \times 128 \times 94$ | $96 \times 96 \times 50$ | $140 \times 140 \times 96$ |
| Voxel size (mm) | $1.5 \times 1.5 \times 1.5$ | $1.98 \times 1.98 \times 2$ | $1.5 \times 1.5 \times 1.5$ |
| Diffusion directions | 42 | 64 | 512 |
| Mask sizes | $88 \times 111 \times 82$ | $61 \times 82 \times 42$ | $84 \times 98 \times 83$ |

Table 2 shows us several insights. First and foremost, we see that the end-to-end processing time of our pipeline is of a *few seconds* at most, even for the large (500 K) trail sets. Given that we bundle a *3D trail set*, and that we perform additional steps during bundling—constrained advection (Section 3.2.1), anisotropy bundling (Section 3.2.2), and tract reseeding (Section 3.2.3), this is just slightly slower than the 2D bundling reported by CUBu [8], which, according to the survey in [7] is by one up to two

orders of magnitude the fastest method for bundling general 2D trail sets. Additionally, note that the bundling step (Section 3.2) is typically done only sparsely during a dataset investigation. Once the desired geometry is obtained, that geometry is next interactively explored using different rendering settings (Section 3.3) and viewpoints. Changing these is, as we mentioned above, real time.

The memory requirements of our method are dominated by storing the density volume needed to sample the KDE density map $\rho$ (Equation (1)). Additional costs involve storing the sample points for $T$. For a dataset of 1M trails, with 100 sample points per trail on average, this involves an extra of roughly 1.2 GB. These costs are not high, given that DTI volumes are typically not acquired at high resolutions, meaning that this acquisition resolution (see Table 1) is an upper bound to that of our density maps. For example, even with 8-fold supersampling of $\rho$ *vs* the actual DTI data (to ensure maximal accuracy for the KDE gradient estimation used in Equation (2), *Dataset 3*, which has $86 \times 116 \times 88$ voxels after cropping the actual brain tissue present in the $140 \times 140 \times 96$ input voxel volume (Table 1), requires a density map of $688 \times 928 \times 704$ voxels, which, if stored with 32-bit-per-voxel density precision, asks for a total of 1.8 GB of GPU RAM. Such figures are definitely within the bounds of modern graphics cards. Hence, from a memory viewpoint, we argue that our method is scalable for the type of data it is aimed at.

**Table 2.** End-to-end execution times of the proposed visualization pipeline for several datasets.

| Dataset | Trail Count $|T|$ | Sample Points | Time/Iteration (ms) | Total Time (ms) |
|---|---|---|---|---|
| Dataset 1 | 50 K | 3.271.979 | 30 | 444 |
| Dataset 2 | 50 K | 1.441.486 | 11 | 170 |
| Dataset 3 | 50 K | 4.145.785 | 20 | 314 |
| Dataset 1 (dense) | 500 K | 32.762.628 | 254 | 3812 |
| Dataset 2 (dense) | 500 K | 12.639.524 | 57 | 858 |
| Dataset 3 (dense) | 500 K | 14.850.241 | 69 | 1029 |

## 5. Internal Evaluation

The bundling-and-rendering pipeline introduced in Section 3 offers a wide set of customization options that allow users to generate a large range of visualizations, as illustrated by the images in that section. However, finding one's way in this visual-design space can be challenging. To address this, we first summarize the free parameters of our method in Table 3. For each parameter, we indicate good preset values we found ourselves by experimenting with its settings. When multiple values are listed for a parameter, e.g., the four profiles $t_F, t_S, t_C, t_G$ for the splat shading (Section 3.3.2), this means that there is no evident best value (preset). For instance, one can use these different profiles to create visualizations that emphasize different, equally interesting, aspects of the data, such as DDH-style rendering (which uses $t_F$) or tube-like rendering with various granularities of the trails (which uses the other three profiles).

**Table 3.** Parameters of the end-to-end DTI bundling-and-rendering pipeline and their presets.

| DTI Bundling | | | Structure-Aware Rendering | | | |
|---|---|---|---|---|---|---|
| CUBu bundling (Section 3.2) | Anisotropic bundling (Section 3.2.2 ) | Tract reseeding (Section 3.2.3) | Shading (Section 3.3.2) | Blending (Section 3.3.2) | Gap filling (Section 3.3.3) | Outlines (Section 3.3.4) |
| $P_R$   13 | $\alpha$   $FA$ | $c_p$   $\geq 0.25$ | $t$   $t_F, t_G, t_S, t_C$ | usage   peel | $r_0$   [3,5] | usage   on, off |
| $P_N$   15 | $\alpha_0$   [0,1] | $c_l$   $\leq 0.2$ | $a$   tangent | $\delta$   0.005 | $\beta$   5 | $\epsilon$   [3,5] |
| $\phi$   0.25 | | | | $\alpha$   0.1 | | |
| $\sigma$   1 | | | | | | |
| $\gamma$   0.2 | | | | | | |

Figure 12 illustrates the above freedom of parameter choice. Here, we select a relatively small set of 65 K trails and render them, both unbundled and bundled, with four styles (blended lines, DDH, shading, and shading plus outlines), using isotropic bundling ($\alpha_0 = 1$). Trails are color-coded on direction. The blended lines style—which corresponds to the classical way DTI trails are rendered in such visualizations—is arguably the most cluttered and does not allow seeing which bundles are in front of others. The DDH style partially alleviates this, but the lack of colors makes visually following bundles hard. Shading offers a good compromise between blended lines and DDH. Finally, shading and outlines make the bundle structures more prominent, but create a more 'loaded' visualization. Separately, we see how isotropic bundling reduces clutter independently on the rendering style, allowing one to peek deeper inside the volume.

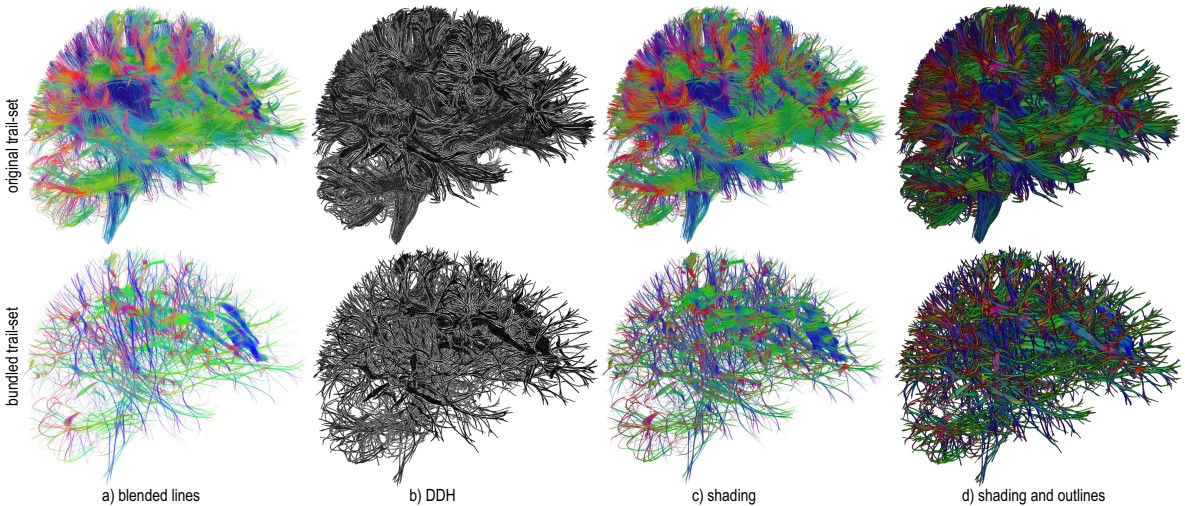

**Figure 12.** Comparing four rendering modes for a small 65 K trail set. Original unbundled data (**top row**) and its bundled counterpart (**bottom row**). See Section 5.

Figure 13 enriches these insights by showing a larger 250 K trail set. Visualizing the unbundled data (top row) creates high amounts of occlusion. The DDH and shading and outlines styles bring in more structure, but cannot, by themselves, reduce occlusion. Isotropic bundling ($\alpha_0 = 1$, middle row) reduces occlusion, as expected, with DDH and shading and outlines bringing in, again, more structure. However, isotropic bundling breaks the tract-sheet structures by creating artificial gaps in these, as discussed in Section 3.2.2. Anisotropic bundling ($\alpha_0 = 0.7$, bottom row) also reduces occlusion but largely preserves sheet structures such as the *corpus callosum* (red). Separately, we see that DDH seems to offer more added value when used on the unbundled data; for the bundled data, the shading and outlines style is able to show structure equally well, and creates easier-to-follow visualizations due to the use of shading.

Concluding this internal evaluation, we found that the parameter defaults listed in Table 3 give good results in terms of creating visualizations with limited occlusion and easy-to-follow visual structures. A few parameters remain free to choose for the user, such as the rendering style and isotropic-vs-anisotropic bundling. We explore these using an external evaluation in the next section.

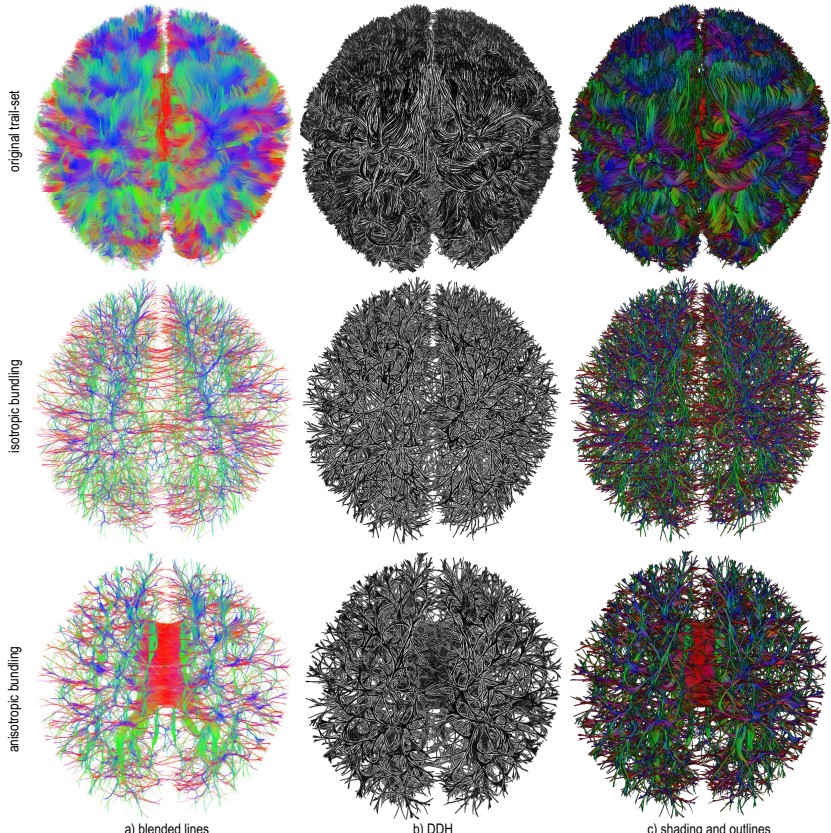

**Figure 13.** Comparing four rendering modes for a large 250 K full-brain trail set. Original unbundled data (**top row**) and its bundled counterpart (**bottom row**). See Section 5.

## 6. External Evaluation

As discussed in Section 5, we found good values for most of the parameters of our bundling-and-rendering DTI visualization pipeline. However, a few parameters cannot be 'frozen' into presets, as they generate widely different visualizations that emphasize different aspects of the data. Apart from that, evaluating our visualization entails also understanding how actual users perceive it; and how they rank the different bundling-and-rendering styles in terms of insightfulness, interestingness, and overall usefulness for understanding the underlying data.

To answer the above questions, we designed an evaluation using external users—that is, persons who are experienced with DTI visualization but who were not involved in the design of our visualization techniques. For the evaluation, we selected a small subset of all possible visualizations that we can obtain by varying the free parameters in Table 3, so as not to overwhelm the users. These visualizations also follow the styles that we found the most effective during our internal evaluation (Section 5), and are described in Table 4 (see also Figure 14). All parameters not specified in Table 4 follow the preset values given in Table 3.

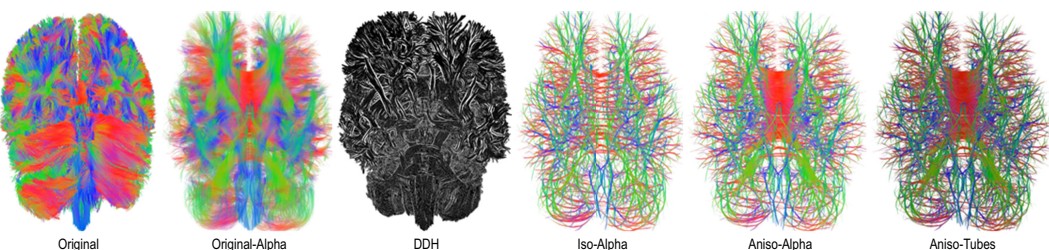

**Figure 14.** Six bundling-and-rendering variations used to evaluate our visualization.

**Table 4.** Six bundling-and-rendering visualization styles used in the external evaluation. See Section 6.

| Style | Description |
| --- | --- |
| Original | No bundling or shading used. Fibers rendered as opaque lines colored by direction |
| Original-Alpha | As Original, but lines are alpha-blended using an opacity $\alpha = 0.1$ |
| DDH | Depth-dependent halo rendering of the unbundled fibers. |
| Iso-Alpha | Isotropic bundling used. Fibers rendered as in Original-Alpha. |
| Aniso-Alpha | Anisotropic bundling used. Fibers rendered as in Original-Alpha. |
| Aniso-Tubes | Anisotropic bundling used. Fibers rendered using the same shading settings as in Figure 12c. |

*6.1. Evaluation Process*

We invited 5 participants with good prior knowledge of DTI data and associated techniques. Their level of experience ranges from 5 to 37 years. One participant is the head of the neuroscience lab in a major French university (37 years experience); one participant is a neuroscientist (12 years experience); two are postdocs with 5 and 10 years experience; and one is a researcher with 10 years experience. Each evaluation lasted one hour and operated as follows.

First, we provided a general introduction of our context and goal. After naming DTI and tractography (which was, as outlined above, well known to the participants), we introduced the problem of data size: The input data for this visualization is a large set (hundreds of thousands) of 3D tracts that follow the main anisotropy directions in a DTI volume. These tracts are in general locally aligned with the directions of the neural fibers. However, directly visualizing a large and densely sampled tract volume creates problems, mainly due to occlusion. We propose to address such problems by a simplified visualization technique that bundles close and same-direction tracts, so that occlusion decreases but the resulting structures still closely reflect the underlying fiber anatomy.

We next introduced the six visualization techniques listed in Table 4 and illustrated them by interactively running our visualization tool. The dataset used during this presentation (and also further on in the evaluation) consists of a brain MRI scan coming from a healthy person, exhibiting no particular anatomical or physiological anomalies. As such, users should expect to see the typical structures they are familiar with from brain anatomy. We traced around 200 K fiber tracts in this brain. The fibers are seeded randomly in regions of high anisotropy (white matter) and traced until they exit this region and enter gray matter, as described in Section 3.1). We explained each of the six visualizations in terms of what the visualization actually *shows*, e.g., what bundling does, what is the working of alpha-blending and shading, and how to interpret the DDH image. During this presentation, we refrained from making *quality* judgments, e.g., state that one visualization can better show certain structures than another one, since that would bias the participants.

Following this global introduction, each user was asked to watch a 3-minute-long video that we prepared in advance. The users could not communicate among themselves, nor with the organizers of the evaluation, during this session, so as not to influence each other. The video shows the DTI trail set described above visualized by the six methods in the order listed in Table 4, i.e., going from the arguably most familiar style to the participants (Original) to the least familiar one (Aniso-Tubes). Each of the six methods is offered roughly the same amount of time in the video. Within such a sequence, the visualization is smoothly rotated along the Y axis, then along the X axis, with the same constant-speed motion patterns for all the sequences. Relevant parameters for the presented visualization style are then varied smoothly, so the participants can understand their effects. For example, for the Alpha styles, we increase the opacity value from the preset $\alpha = 0.1$ to reach 1 (full opacity) and then back, so one understands how the alpha-blending works. To understand the bundled styles, the video varies the relaxation parameter $\gamma$ between the preset value ($\gamma = 0.2$, strongly bundled) to 1 (fully unbundled) and then back. Each sequence is captioned in the video by its name. Additional captions explain the parameters that are varied. Figure 15 shows eight frames from the watched video.

Using a video instead of directly allowing users to interact with the visualization is, of course, limited, as one cannot change the visualization parameters at will, nor interactively manipulate the viewpoint. However, a video also has important advantages: (1) It offers exactly the same information to all participants, therefore decreasing potential bias due to participants using different settings or options. (2) Letting participants actually interact with the visualization tool presupposes a fine-tuned GUI design, tool manual, and documentation, all of which are out of our evaluation scope. (3) Using a fixed-length video, with a pre-recorded narrative, allows one to time-box the experiment far easier than if users were allowed to freely explore all options of the visualization tool. Given these, we opted for the video presentation variant.

The users could watch the video at any desired pace, including going back to watch again some sections. During this time, users were also asked to answer several questions that compare the various pros and cons of the six visualization methods. Besides providing these answers, users also could provide free-text comments. The full questionnaire is available in Appendix A. The questionnaire contains three parts: (1) Listing the strongest points and limitations of each of the six visualizations (free text input); (2) Relatively ranking the six visualizations concerning their ability to handle occlusion, showing brain structures clearly, and showing how various brain regions are connected by these structures (ordinal scale from 1 = best to 6 = worst); (3) Global aspects, including naming the overall preferred visualization, the perceived advantages and limitations of bundling, and aspects present in different visualization styles that deserve to be combined. Users could start filling the questionnaire during the watching or the video, or afterwards, as desired. The total time given for watching the video and filling the questionnaire was approximately 40 min.

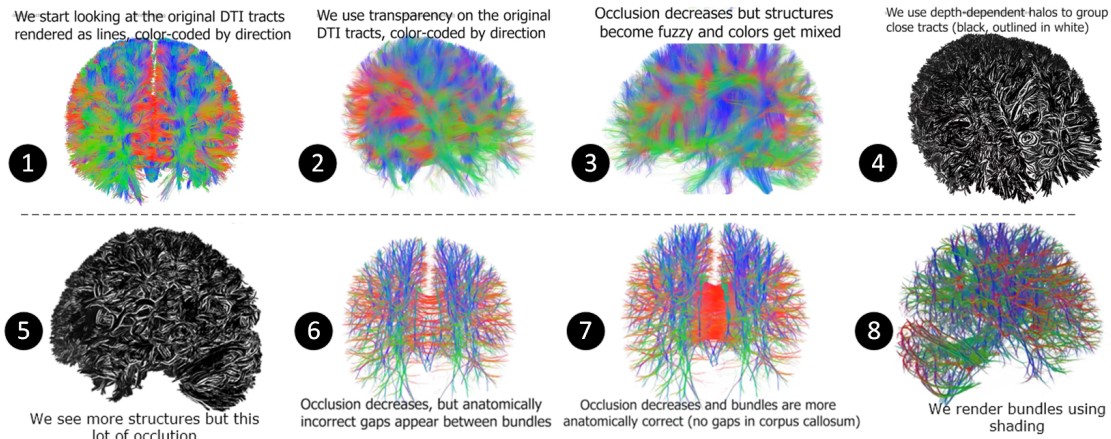

**Figure 15.** During our evaluation, we show users a video that depicts the six visualization styles (Table 4) using a 3D animation that rotates the viewpoint to see the brain from multiple angles. This image shows 8 snapshots from the video. Captions inform the user of the specifics of the visualization style being shown.

*6.2. Evaluation Summary*

We summarize below the strengths and limitations of the six analyzed visualization styles as reported by the participants.

**Original:** The superficial layers are clearly visible, which helps to figure out error in the fiber reconstruction (tracking). This visualization is a classical view that is close to the raw data provided by the machine. On the negative side, too many occlusions severely hinder the visualization of the fibers within the model. The depth cannot be assessed, and the inner structure is not visible.

**Original-Alpha:** This style provides a fairly good visualization of large and dense fiber groups with an improved possible assessment of the fiber depth. The inner structures are visible as long as they correspond to such large and relatively dense fiber structures. On the negative side, the view is too

cluttered, and many groups of fibers are hard to distinguish. The inner part of the model is still difficult to get insights in. Moreover, blending makes it hard to distinguish how overlapping bundles cross (what is in front and what behind) from a single viewpoint.

**DDS:** This visualization was qualified to be an 'aesthetic representation' of the fibers. Some structures on the model's peripheral layer (outer core) are better visible with DDH than the other visualizations. On the negative side, neurologists are used to interpret grayscale imagery in specific ways—e.g., as encoding scalar quantities—whereby black and white have specific meanings. DDH's usage of black and white can be confusing at moments. Also, the restriction to black and white removes the possibility of color-coding directional information, which is often needed. DDH also shows a lot of occlusion. Overall, DDH has been assessed as too complex and making small fibers not visible anymore.

**Iso-Alpha:** This visualization shows well the directions of the relevant (main) fibers with a good depth understanding; the fiber paths are clearly visible. On the negative side, this view shows unrealistic fibers path with too strong distortions. The user needs to recall the original fiber structure to interpret the displayed information.

**Aniso-Alpha:** In this visualization, the fibers that form the *corpus callosum* are correctly aggregated into a sheet. Overall, Aniso-Alpha creates a more realistic brain-fiber visualization that depicts the brain structure quite well. On the negative side, Aniso-Alpha does not show individual fibers and can create a 'hairy' representation. The spinal cord is not correctly rendered (too many fibers are still spread).

**Aniso-Tube:** The depth perception is improved leading to a better understanding of the aggregated fiber structure. Outlines of the bundles are better visible than in the other visualizations, due to the shading; this also emphasizes the fibers' directions. The visualization is also realistic in terms of the depicted brain-fiber structures. On the negative side, the view remains cluttered with many small bundles; the global opacity of the bundles makes it hard to see deep(er) inside the fiber set.

*6.3. Ranking of the Visualizations*

As outlined in Section 6.1, we also asked the participants to rank the six studied styles regarding their effectiveness for occlusion removal, clarity and brain connectivity understanding. The summarized user answers are listed below, with a focus on the extremes—top ranks (1, 2) and bottom ranks (5, 6):

1. *Occlusion removal:* Aniso-Tubes is ranked as first or second best; DDH and Origin are ranked last or last but one;
2. *Clarity:* Aniso-Tube and Aniso-Alpha are ranked first or second in terms of clarity. Iso-Alpha and DDH are ranked last and last but one;
3. *Connectivity:* Aniso-Alpha and Aniso-Tubes are ranked first or second best; DDH is ranked last or last but one for the same task.

*6.4. General Comments*

We next summarize the textual answers that the users gave to questionnaire point 3. Overall, the users liked Aniso-Tube most, which they found to provide a good compromise between view aggregation and general structure understanding thanks to the sharp-rendered fiber borders due to shading. This style was also found to preserve the general brain structure well while clearly showing the main bundles and the brain connectivity. Although it was acknowledged that this style is affected by the inherent distortion caused by bundling, the distortion was (correctly) found to be less than the Iso styles (which, indeed, distort both in planar anisotropy and linear anisotropy areas).

The depth perception was also found to be good, and helped by the shading's ability to accentuate bundle borders. Interestingly, participants also mentioned combining DDS with Aniso-Tube to further enhance the bundle border cues. This is salient, since the participants were *not aware* of the shading and outlines style (Figure 13), which does exactly that, as that style was not part of the evaluation video. This represents, we believe, good support for the potential added value of shading and outlines.

## 7. Discussion

We discuss next briefly the main characteristics of, and findings related to, our proposed method.

**Technical points:** All steps of the presented method are generic, in the sense that they can be applied to any trail set consisting of a set of 3D point-sampled trajectories that is further annotated with anisotropy scalar values. A subset of the steps (3D bundling, shading, outline computation) are actually more generic in the sense that they only require a set of 3D point-sampled trails. The entire method is implemented on the GPU, and has a complexity linear in the number of sample points $\mathbf{x}_j$. Practically, this allows us to bundle-and-render hundreds of thousands of 3D trails at interactive frame rates.

**Quality:** The quality of the produced visualizations, in terms of ability to recover complex brain structures such as fiber sheets, depends strongly on the availability of a *dense* trail set that samples the DTI volume. Indeed, a too sparse trail set will not be able to capture these structures, so bundling and shading can do little to improve this. Reseeding (Section 3.2.3) alleviates this by creating additional tracks on-the-fly during the visualization. However, this cannot *guarantee* a dense sampling of all regions of interest. As such, starting with a rich, densely sampled, trail set is an important condition for quality.

**Veracity:** A well-known problem of bundling methods is that they deform the actual data, trading off veracity for simplification [7]. Although this is less critical when bundling abstract data, such as graph drawings, deforming DTI fibers is more problematic, as these can potentially convey false insights to the user. The anisotropic bundling techniques described in Sections 3.2.1 and 3.2.2 limit such problems as they *constrain* the bundling to follow additional data-related information (anisotropy). Still, the inherent trade-off of bundling between simplification and veracity remains. This is not a problem specific to our method—many other methods that produce simplified visualizations of DTI data share the same issue [14,16,60]. Exploring how to both *measure* and *limit* deformations caused by bundling that create false insights is an interesting and important direction for future work.

**Evaluation:** Our evaluation is inherently limited by the format chosen—inviting only professionals deeply familiar with DTI visualization, exposing them to precisely the same material, and setting a time limit of under one hour for the entire process. Inviting more people (with less DTI experience), allowing them to actually experiment live with the visualization tool, and recording more and more task-specific information would, obviously, give more insights into the specific strengths and weaknesses of our method. However, we believe that this type of targeted evaluation can only follow our *formative* evaluation. Indeed, the latter could have delivered evidence that does not support the overall added-value claim for our visualization, in which case doing a targeted evaluation would not have made sense. The results of our evaluation, however, have singled out Aniso-Tubes and Aniso-Alpha as the top preferred styles for occlusion removal, clarity, and depicting connectivity. Hence, we have evidence that *anisotropic bundling* (present only in the Aniso style) and *shading* (present only in the Tubes style) are both of added value. With this information, we plan to execute a subsequent evaluation—using the combination of parameters mentioned above as presets—in which several more precisely defined tasks, with ground-truth information available for measuring the quality of completion, will be addressed using our tool.

## 8. Conclusions

In this paper, we have presented a set of techniques for the simplified visualization of large 3D trail sets produced by tractography on DTI volume data. We perform simplification jointly both in the geometric space and image space. For the geometric simplification, we extended the state-of-the-art CUBu bundling method to handle 3D trail sets, and also proposed several bundling adaptations to handle the specifics of DTI fiber tracks. For the image-space simplification, we proposed several splat-based rendering methods that merge the rendered trails in the resulting image to create more compact sheet-like structures, add shading, and delineate bundles by outlines. The geometric and image-space simplifications can be used jointly but also independently. We implemented the entire pipeline on the GPU using a combination of CUDA and pixel shader techniques. Our end-to-end method can create simplified visualizations at interactive frame rates and allow interactive parameter manipulation.

We studied the parameter space of the proposed method by identifying good presets for most of them. For the remaining free ones, we explored their various combinations by an internal and an external evaluation, the latter involving five professionals working with DTI data. The combined evaluations show that both bundling and shading have perceived added value and are seen as creating images that have less occlusion, are clearer, and show the brain connectivity better than other comparable techniques.

Several directions of future work are possible. First, we aim to further explore the design space spanned by our various bundling and shading techniques, so as to find different combinations of these and/or their parameters that lead to visualizations with better occlusion reduction, clarity, and ability to depict complex brain structures. Secondly, we consider adapting our pipeline to incorporate (and measure) quality criteria to show and control the level of permissible deformation caused by bundling, thereby increasing the confidence of users in the depicted structures. Last, a low-hanging fruit is to use our pipeline to create simplified visualizations of other large 3D trail sets such as aircraft trajectories, fluid flow data, or graph layouts.

**Author Contributions:** S.B. was responsible for the implementation and testing of the presented set of techniques (Section 3). C.H. was responsible for executing the user evaluation (Sections 5 and 6). A.T. was responsible for the overall research ideas, methodology, and formalization of the reported results. All authors have read and agreed to the published version of the manuscript.

**Funding:** This research received no external funding.

**Conflicts of Interest:** The authors declare no conflict of interest.

## Appendix A. Evaluation Questionnaire

In the following, we list the questionnaire that was used for the external evaluation described in Section 6.

**Part 1: Merits and limitations of the six visualizations**

---

**Original**
The strong points of this visualization are [ *free text* ]
The weak points of this visualization are [ *free text* ]

---

**Original-Alpha**
The strong points of this visualization are [ *free text* ]
The weak points of this visualization are [ *free text* ]

---

**DDH**
The strong points of this visualization are [ *free text* ]
The weak points of this visualization are [ *free text* ]

---

**Iso-Alpha**
The strong points of this visualization are [ *free text* ]
The weak points of this visualization are [ *free text* ]

---

**Aniso-Alpha**
The strong points of this visualization are [ *free text* ]
The weak points of this visualization are [ *free text* ]

---

**Aniso-Tubes**
The strong points of this visualization are [ *free text* ]
The weak points of this visualization are [ *free text* ]

---

**Part 2: Relative ranking of the six visualizations**

---

**Occlusion**
Rank the six visualizations on a scale 1 to 6 (1 = best, 6 = worst) in terms of how well they allow you to see structures which are deep inside the volume. You cannot give the same score to two different visualizations.
Original [ ] Original-Alpha [ ] DDH [ ] Iso-Alpha [ ] Aniso-Alpha [ ] Aniso-Tubes [ ]

---

**Clarity**
Rank the six visualizations on a scale 1 to 6 (1 = best, 6 = worst) in terms of how easy is to discern the structures they depict. You cannot give the same score to two different visualizations.
Original [ ] Original-Alpha [ ] DDH [ ] Iso-Alpha [ ] Aniso-Alpha [ ] Aniso-Tubes [ ]

---

**Connectivity**
Rank the six visualizations on a scale 1 to 6 (1 = best, 6 = worst) in terms of how well they allow you to see how two zones in the brain are connected (or not). You cannot give the same score to two different visualizations.
Original [ ] Original-Alpha [ ] DDH [ ] Iso-Alpha [ ] Aniso-Alpha [ ] Aniso-Tubes [ ]

---

**Part 3: Open comments**

Overall, considering all tasks mentioned in part two, but also other tasks typical in DTI visualization, which of the six visualizations do you prefer, and why?

[ *free text* ]

Are there any aspects you see in different visualizations that you would like to see combined in a single visualization? If so, which ones?

[ *free text* ]

Considering all above aspects, which is the strongest added value you see for bundling in general, and for anisotropic bundling in particular?

[ *free text* ]

Considering all above aspects, which is the strongest limitation you see for bundling in general, and for anisotropic bundling in particular?

[ *free text* ]

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
