# Peer review of "Structure-Aware Trail Bundling for Large DTI Datasets"

_algorithms, doi:10.3390/a13120316_

Round 1

Reviewer 1 Report

The paper presents a flexible and efficient 3D bundling strategy for brain fiber tracking, visualization and simplification. The main contribution is the extension to 3d of a 2D fiber bundling algorithm and the improvement in processing time thus contributing to the scalability of the approach.

A rudimentary user evaluation was performed, which, although limited, does not affect the value of the contribution.

Two important issues remain that I believe should be handled by the authors before publication. They have to do with related work and writing/presentation.

1 - Related work. The authors chose to generally give directions on  DTI imaging and on trail and bundling. The second part is adequate to the work, but I missed a discussion of current visualization techniques for brain fiber tracking, given the focus on that particular application.  It would be reasonable to expect their presentation of such techniques and also to compare in what way their visualizations differ. the Scientific soundness rating is in regards to that aspect of the contextualization of the work. I have not detected any inadequacy otherwise. I mention just a few references found in a quick search, in regards to work in the visualization of 3D brain fibers:

Xu, C., Liu, YP., Jiang, Z. et al. Visual interactive exploration and clustering of brain fiber tracts. J Vis 23, 491–506 (2020). https://doi.org/10.1007/s12650-020-00642-1

Yeatman, J.D., Richie-Halford, A., Smith, J.K. et al. A browser-based tool for visualization and analysis of diffusion MRI data. Nat Commun 9, 940 (2018). https://doi.org/10.1038/s41467-018-03297-7

Poco, J., Eler, D.M., Paulovich, F.V. and Minghim, R. (2012), Employing 2D Projections for Fast Visual Exploration of Large Fiber Tracking Data. Computer Graphics Forum, 31: 1075-1084. https://doi.org/10.1111/j.1467-8659.2012.03100.x

2 - Adjustments in writing and presentation. Although the paper is generally written in good English and it is easy to understand in terms of language, adjustments should be made.

2.1  In many places, particularly towards the end of the document, an additional thorough review of language should be performed. I am going to give some examples, but there are really many more.

Examples:

 - the use of which, such as in 

"3D DTI tracts which allow...", it is either "3D DTI tracts that allow ", or 3D " DTI tracts, which allows"; which is preceded by a comma. 

and

" widely different  visualizations which emphasize differently" 

I gave 2 examples, but it happens in various other parts of the text.

-  we introduce first some notations  ->  we first introduce?

- ... a 3D volume scan that encoders, per voxel,...   ->  encodes?

- lines 151-152

" First, we densely seed M to start tracing streamlines in the vector
152 field e1 from points inside it. Secondly,... "  -> First and second or Firstly and secondly?

- line 416/417 "  is arguably the most cluttered as does not allow " ->   as - it-  does not allow?

- there is a trail set and trail-set in the text.

Line 390:  " Summarizing, we claim that our method offers interactive response rates to users interested..."  That sentence does not summarize the content of the paragraph. Additionally, at this point, you are probably demonstrating, not claiming...

2.2 The abstract and the introduction should summarize the methodology and the results. When mentioning "and produces visualizations which have been found as better in occlusion reduction, conveying the connectivity structure of the brain, and overall
clarity than existing methods for the same data", it should mention how that was ' found'.  Are these aspects measured numerically?  Is it user opinion? In other words, try to be more precise in terms of results. Otherwise, the reader has to find the details inside the later section to understand how that conclusion was reached.

2.3 In some places the writing seems to shorten the description too much. for instance, in the sentence:

" Besides
their spatial position, trails can be annotated by additional data attributes, such as edge weights, directions, and types (graphs), or type, speed, and travel duration (motion datasets). "   

The first (graph)  is a representation. The second (motion)  is an application of that representation.  It does not seem to make sense in the context of that paragraph. Please clarify.  This first paragraph sets the framework in which the results of the paper are going to be inserted. It is important to have a better explanation.

2.4 In the abstract you claim the algorithm has only a few parameters with a simple meaning. However, you start Sec. 5 by saying:

" The bundling and rendering pipeline introduced in Sec. 3 offers a wide set of customization options that allow users to generate a large range of visualizations, as illustrated by the images in that section. However, finding one’s way in this visual-design space can be challenging. To address this, we first summarize the free parameters of our method in Tab. 3." 

If it is challenging and you need a table to summarize, maybe the parameters are not so few or that simple?

2.5 Please make it more clear the precise between the current method and the previous contribution Cubu.  Preferably in Related works.

These are examples of writing, presentation, and contextualization issues that occur all through the text. I believe a thorough review would bring the authors to these same observations and should work to improve the document.

Reviewer 2 Report

I found the work quite impressive, particularly that it covers all stages of the pipeline (from tract generation to budling, to rendering) and proposes innovative solutions and tweaks in each one (e.g., anisotropic bundling).

The evaluation with domain experts was appreciated though somewhat minimal. While they were able to comment on the aesthetics of the representation (with an expert's eye) it is unclear whether the novel representations fit well with the type of interactions and analytical tasks they are doing (a few problems are alluded to: difficulty assessing 'broken tracts'). I think this is a minor point though. The contributions of the paper as is are significant and my intuition that the visualizations would be fine in an applied context too.

Overall I think the work is publishable and meaningful as is.
